# The brain structure, inflammatory, and genetic mechanisms mediate the association between physical frailty and depression

Rongtao Jiang [1] ✉, Stephanie Noble [2,3,4], Matthew Rosenblatt [5], Wei Dai [6], Jean Ye [7], Shu Liu [8,9], Shile Qi [10], Vince D. Calhoun [10], Jing Sui [11] ✉ & Dustin Scheinost [1,5,7,12,13]

Cross-sectional studies have demonstrated strong associations between physical frailty and depression. However, the evidence from prospective studies is limited. Here, we analyze data of 352,277 participants from UK Biobank with 12.25-year follow-up. Compared with non-frail individuals, pre-frail and frail individuals have increased risk for incident depression independent of many putative confounds. Altogether, pre-frail and frail individuals account for 20.58% and 13.16% of depression cases by population attributable fraction analyses. Higher risks are observed in males and individuals younger than 65 years than their counterparts. Mendelian randomization analyses support a potential causal effect of frailty on depression. Associations are also observed between inflammatory markers, brain volumes, and incident depression. Moreover, these regional brain volumes and three inflammatory markers—C-reactive protein, neutrophils, and leukocytes—significantly mediate associations between frailty and depression. Given the scarcity of curative treatment for depression and the high disease burden, identifying potential modifiable risk factors of depression, such as frailty, is needed.

Depression is a leading cause of disability worldwide[1]. Due to the scarcity of curative treatment and the high disease burden, identifying modifiable risk factors that prevent depression constitutes an urgent public health priority[2,3]. Many modifiable factors have been explored as components of prevention strategies, including increased physical activity, decreased daytime napping, and reduced sedentary behaviors[4–6].

Recently, physical frailty has been attracting growing attention as a potential target due to its pervasiveness and strong associations with multiple health outcomes, including depression[7,8]. Frailty describes a state of increased vulnerability to stressors resulting from decreased physiological reserve[9], and may be improved with targeted intervention. Evidence from recent studies consistently suggests more severe depressive symptoms[7,10] and a 2.64-fold increased risk of developing depression for frail people[11]. This body of research offers important preliminary insights into the link between frailty and depression. However, several critical knowledge gaps remain. First, evidence from

[1]Department of Radiology and Biomedical Imaging, Yale School of Medicine, New Haven, CT 06510, USA. [2]Department of Psychology, Northeastern University, Boston, MA, USA. [3]Department of Bioengineering, Northeastern University, Boston, MA, USA. [4]Center for Cognitive and Brain Health, Northeastern University, Boston, MA, USA. [5]Department of Biomedical Engineering, Yale University, New Haven, CT 06520, USA. [6]Department of Biostatistics, Yale University, New Haven, CT 06520, USA. [7]Interdepartmental Neuroscience Program, Yale University, New Haven, CT 06520, USA. [8]Department of Psychiatry, Amsterdam UMC, University of Amsterdam, Amsterdam, the Netherlands. [9]Amsterdam Neuroscience, Amsterdam, the Netherlands. [10]Tri-institutional Center for Translational Research in Neuroimaging and Data Science (TReNDS), Georgia State University, Georgia Institute of Technology, and Emory University, Atlanta, GA 30303, USA. [11]State Key Laboratory of Cognitive Neuroscience and Learning, Beijing Normal University, Beijing, China. [12]Department of Statistics & Data Science, Yale University, New Haven, CT 06520, USA. [13]Child Study Center, Yale School of Medicine, New Haven, CT 06510, USA. ✉e-mail: rongtao.jiang@yale.edu; jsui@bnu.edu.cn

previous studies was either cross-sectional or had a short follow-up period, raising the concern that physical frailty may be a prodromal syndrome preceding the occurrence of depression rather than a risk factor[12]. Second, previous studies did not adequately account for many confounds occurring with frailty and depression, including metabolic syndrome and unhealthy lifestyles, which may obscure the true associations. Furthermore, observational studies cannot draw causal inferences due to residual confounding and potential reverse causality. Only one recent study has examined the causal relationship between frailty and depression[13], but they assessed frailty using the health deficit accumulation approach, which is less modifiable than the frailty phenotype[9]. Third, most studies have relied on small samples characterized by elderly-oriented settings. Although frailty was initially studied in older people and increases with age, it can also be identified in younger people[14]. Moreover, the mechanisms by which physical frailty might predispose an individual to depression are not fully understood. There is accumulating evidence that the immunoinflammatory system and brain structure are disturbed in people with frailty and constitute key elements in the pathophysiology of depression[7,15–18]. Yet existing data regarding whether frailty-related inflammatory markers or brain changes contribute to frailty-depression associations remains scarce. Therefore, a comprehensive study with multidimensional, longitudinal data is needed to understand associations between frailty and depression incidence and potential mechanisms.

Leveraging data from over 350,000 participants from the UK Biobank, this study investigated the prospective association between physical frailty and incident depression and revealed an increased risk of developing depression in people with pre-frailty and frailty. Mendelian randomization (MR) analysis provided further support for a potential causal effect of frailty on depression. We also explored the potential mechanisms driven by inflammatory markers and brain structure and found that subcortical brain structures and inflammatory markers significantly mediated the association between frailty and depression. Overall, our results highlighted the detrimental effect of physical frailty on depression and elucidated the potential mechanisms driven by brain structure and systemic inflammation.

## Results

### Study sample

This study used data from UK Biobank, a population-based cohort study of over 500,000 participants aged 37–73. We adopted the physical frailty phenotype to assess frailty severity[19], which included the following five indicators: weight loss, exhaustion, weakness, physical inactivity, and slow walking speed. Three mutually exclusive groups of frailty, pre-frailty, and non-frailty were defined for participants fulfilling three or more, one or two, or no criteria, respectively. Figure 1 shows an overview of analyses performed in the current study.

A total of 352,277 participants were included in the final analyses, of whom 11,241 met the criteria for frailty, 138,111 for pre-frailty, and 202,925 for non-frailty. Participants were 51.77% female, 95.15% white, and had a mean age of 56.48 years. Baseline characteristics by frailty status are detailed in Supplementary Table 1 and Supplementary Fig. 1. Overall, frailty and pre-frailty are associated with female sex, material deprivation, lower educational attainment, lower income, and a higher prevalence of metabolic syndrome. Also, frail and pre-frail participants are more likely to smoke and spend more time in sedentary behaviors but drink alcohol less frequently than non-frail individuals.

### Prospective association between physical frailty and depression incidence

During a median follow-up of 12.25 years (IQR 11.52–12.94 years), 11,269 depression cases were documented. Compared with non-frail individuals, those with pre-frailty and frailty had a significantly increased risk for depression incidence. The risk was 1.60-times higher for pre-frailty (HR = 1.60, 95% CI = [1.53–1.66]) and 3.20-times higher for frailty (HR = 3.20, [2.98–3.43], all Bonferroni corrected $P < 0.001$) after adjustment for covariates including age, sex, race, alcohol intake, smoking status, sedentary behavior, education level, material deprivation, family income, and metabolic syndrome (Fig. 2A). Pre-frailty and frailty have higher HRs than any covariates included in the model (Supplementary Table 2). These results did not change appreciably when using a 10-year landmark analysis or including an extended set of covariates (Supplementary Fig. 2, Supplementary Table 3). The risk for individuals presenting one to five frailty criteria was 1.45-, 2.11-, 3.09-, 3.79-, and 4.37-fold higher than those presenting none, respectively

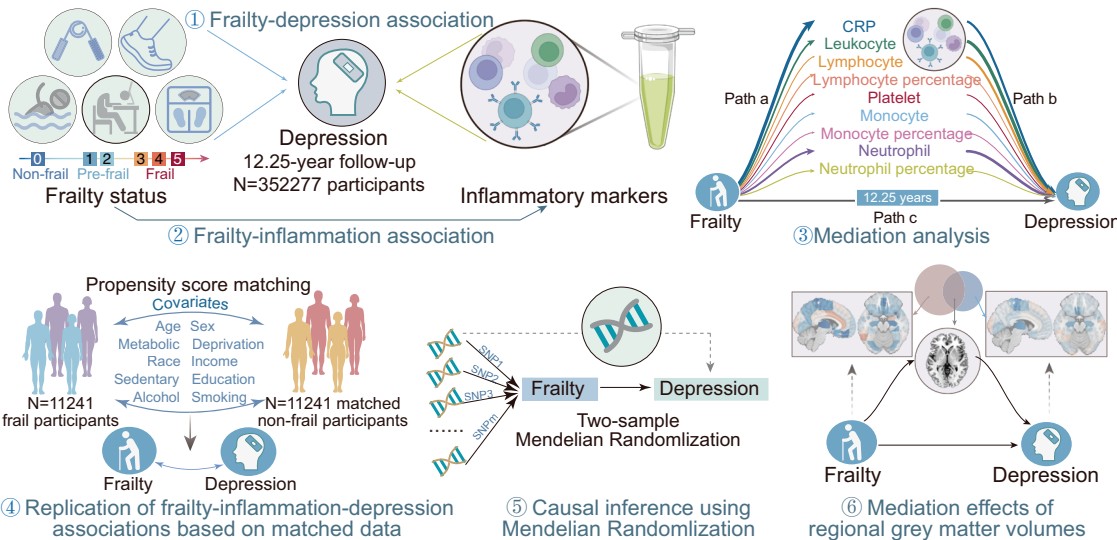

**Fig. 1 | Overview of analyses performed in the current study.** Leveraging data from the UK Biobank, this study investigates the prospective association of physical frailty and nine inflammatory markers with incident depression, and the mediating effects of inflammatory markers on the association between physical frailty and incident depression. These analyses are subsequently replicated based on matched frail and non-frail samples using a propensity score matching procedure to validate the main results. Then, this study uses Mendelian randomization to make causal inferences of the effect of physical frailty on depression. Finally, this study investigates how physical frailty relates to regional gray matter volume and further examines the mediating effect of these regional gray matter volumes on the association between physical frailty and depressive symptoms. CRP, C-reactive protein. Icons were made from https://www.svgrepo.com/.

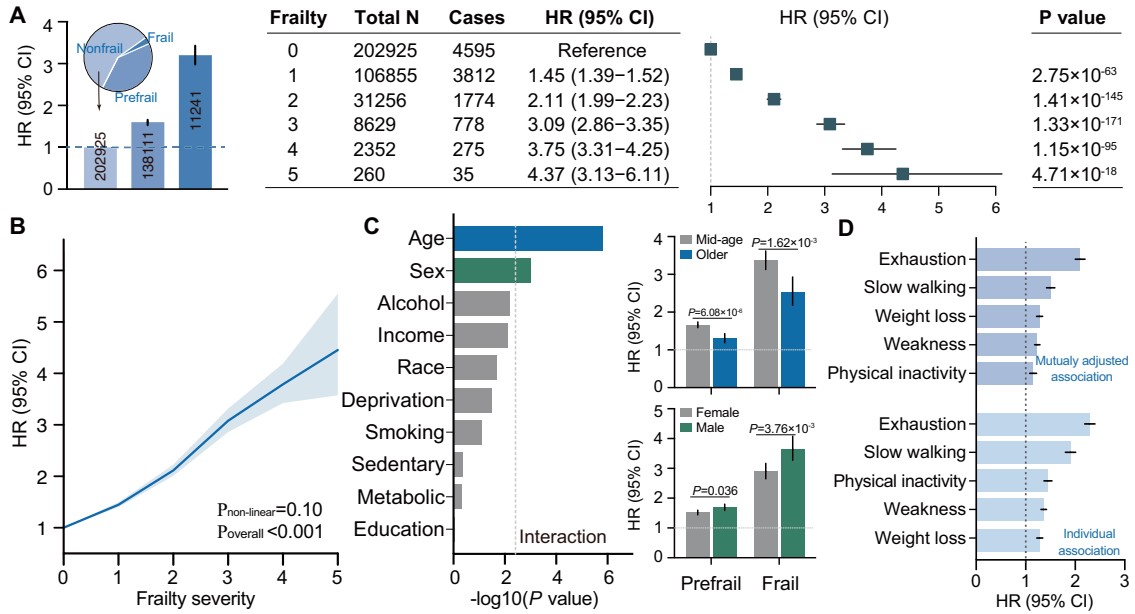

**Fig. 2 | The prospective associations between physical frailty and incident depression. A** Compared with non-frail individuals, those with pre-frailty and frailty were at higher risk of developing depression after adjustment for covariates. When physical frailty was modeled as an ordinal variable, the depression risk increased along with the number of frailty indicators. **B** No evidence of non-linearity was supported. **C** Among all confounds, only age and sex showed significant interaction with physical frailty on depression incidence, with more pronounced effects observed in males and middle-aged people than in females and older people, respectively. The dashed vertical line indicates threshold of significance after correcting for multiple testing ($-\log(0.05/10) = 2.30$). Frail males: $n = 4391$; frail females: 6850; pre-frail males: $n = 62,158$; pre-frail females: $n = 75,953$; middle-aged pre-frail: $n = 108,998$; middle-aged frail: $n = 8476$; older pre-frail: $n = 29,113$; older frail: $n = 2765$. Size of the bar represents -log10(P value). **D** Each of the five components of physical frailty showed independent associations with the risk of depression incidence, even in mutually adjusting models where all five components are simultaneously modeled as exposures in Cox hazard models ($n = 352,277$). Two-sided unadjusted P values and HRs were derived from Cox proportional hazard model (Z-tests). For all figures, the size of the bars and the internal center represent mean Z-HRs. The error bar and horizontal lines indicate the corresponding 95% CI. All P values were two-sided, and unadjusted. HR hazard ratio. Source data are provided as a Source Data file.

(Fig. 2A). The exposure-response curve between physical frailty and incident depression is shown in Fig. 2B, and no evidence of non-linearity was observed ($P = 0.10$). Strong evidence also supported the interaction between physical frailty and age ($P = 1.49 \times 10^{-6}$) and physical frailty and sex ($P = 9.68 \times 10^{-4}$), but not for the other covariates (Bonferroni-corrected P value $< 0.05/10$ for 10 tests, Fig. 2C). In stratified analyses, the associations between frailty and depression tend to be higher in males ($HR_{pre-frailty} = 1.69$, $HR_{frailty} = 3.66$) compared with females ($HR_{pre-frailty} = 1.52$, $HR_{frailty} = 2.90$), and in middle-aged adults ($HR_{pre-frailty} = 1.66$, $HR_{frailty} = 3.37$) compared with their older counterparts ($HR_{pre-frailty} = 1.30$, $HR_{frailty} = 2.53$, Supplementary Table 4).

Each of the five components of physical frailty showed independent associations with the risk of depression incidence (HR ranged from 1.15 to 2.10, Supplementary Table 5). When the analyses were mutually adjusted by components of frailty, the HRs were slightly attenuated but remained significant (Fig. 2D). Specifically, exhaustion (HR = 2.10, [2.00, 2.20]) demonstrated the strongest association with depression incidence, while physical inactivity showed the least significant association (HR = 1.15, [1.08, 1.22]). Based on population attributable fraction analyses, pre-frailty and frailty accounted for 20.58% and 13.16% of depression cases. Among the five components of frailty, exhaustion (13.07%) had the highest population attributable fraction, and physical inactivity (4.04%) had the lowest fraction.

### Causal inference of the effect of physical frailty on depression

We performed MR analysis for causal inference between genetically predicted physical frailty and the risk of depression. The MR-Egger intercept term indicated no obvious directional pleiotropy (intercept=0.003, $P = 0.810$), but Cochran's Q test revealed significant heterogeneity (Q = 79.89, $P = 1.20 \times 10^{-6}$). Thus, the inverse-variance weighted (IVW) method under multiplicative random effect was used

as the primary method[13]. Using 30 significantly frailty-associated SNPs (single nucleotide polymorphisms, Supplementary Table 6) as proxies, the IVW method found that each one-point increment in physical frailty was associated with a 2.55-times higher risk of depression (odds ratio (OR) = 2.55, 95% CI= [1.59, 4.08], $P = 1.01 \times 10^{-4}$, Supplementary Fig. 3). The MR estimates based on the weighted median (OR = 1.92, 95% CI = [1.19, 3.10], $P = 7.96 \times 10^{-3}$) and MR-Egger (OR = 2.02, 95% CI= [0.29, 13.95], $P = 0.482$) showed consistent detrimental effects of physical frailty on depression, but were only significant for the weighted median. MR-PRESSO detected three outliers, the removal of which had a nominal impact on effect estimates (OR = 2.73, 95% CI= [1.82, 4.10], $P = 1.37 \times 10^{-6}$). Further, leave-one-SNP-out analyses suggest that no single SNP drove these MR estimates (Supplementary Fig. 3). Sensitivity analysis under a relaxed threshold for selecting significant instruments ($P < 5 \times 10^{-7}$) yielded similar results as the primary finding (Supplementary Fig. 4), providing further evidence for the stability of the causal inferences.

### Associations between physical frailty and inflammatory markers

All nine inflammatory markers showed significant associations with physical frailty while controlling for covariates and multiple comparisons (Fig. 3A and Supplementary Fig. 5). The severity of frailty was negatively correlated with the lymphocyte percentage and monocyte percentage and positively correlated with the other seven markers (Supplementary Table 7). The strongest effect was observed for serum CRP (d = 0.24, Fig. 3B).

### Prospective association between inflammatory markers and depression incidence

Strong associations were observed between eight of the nine inflammatory markers (except monocyte percentage) and incident

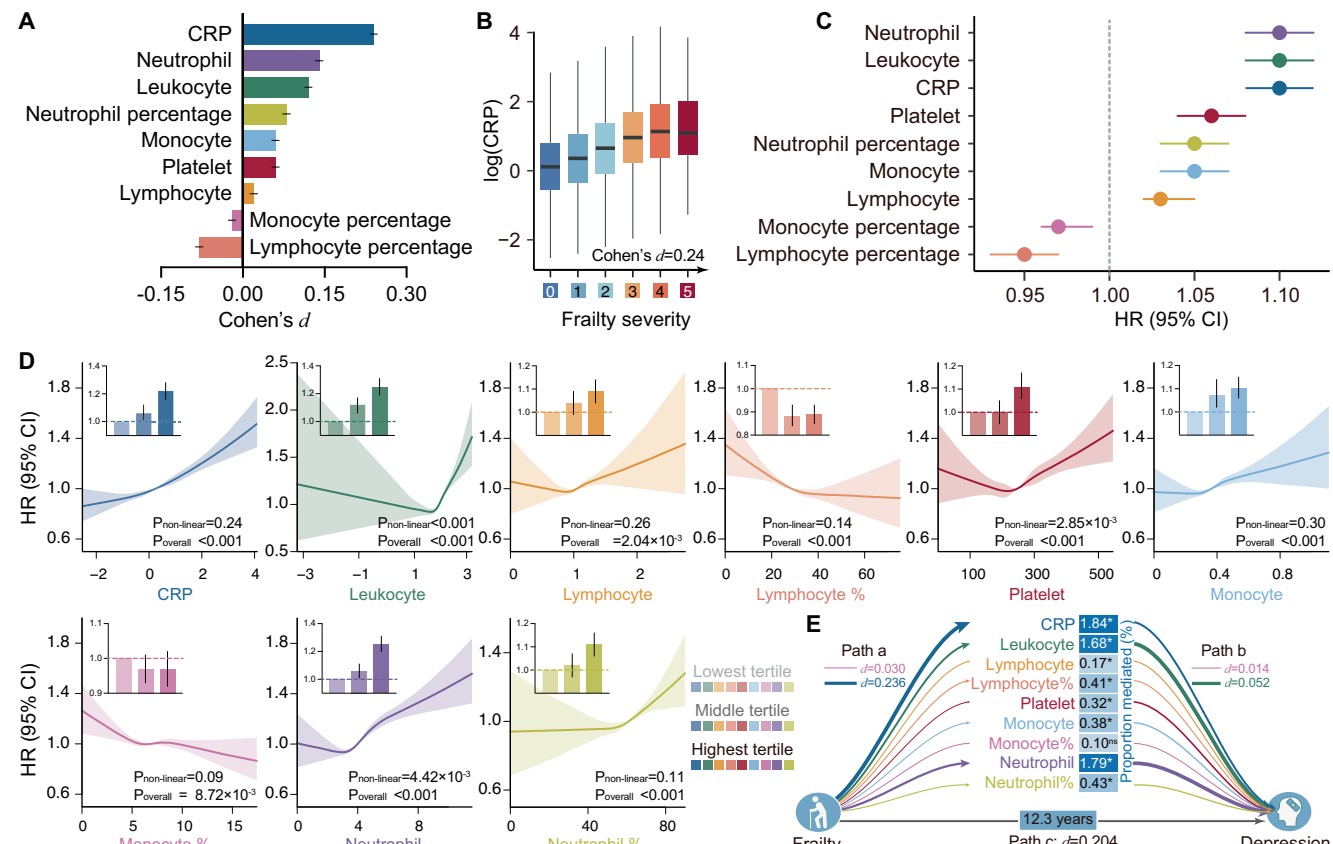

**Fig. 3 | Prospective association of inflammatory markers with physical frailty and depression incidence. A** All nine inflammatory markers showed significant associations with physical frailty while controlling for several covariates. Analysis employed linear mixed-effects models, which tested the statistical significance of coefficients against a t-distribution. Size of the bar and the internal center represent mean Cohen's d. The error bar represents the corresponding 95% CI. **B** Serum CRP showed the strongest association with physical frailty ($n = 352,277$). Boxplot elements were defined as follows: the center line is the median value; box limits are the upper and lower quartiles; whiskers are 1.5× the interquartile range. **C** Cox proportional hazards models provided evidence of linear associations between eight of the nine inflammatory markers (except monocyte percentage) and incident depression (Z-tests). Dots: mean HRs; Horizontal lines: 95% CI. **D** The exposure-response curve between inflammatory markers and risk of depression incidence. Significant non-linear associations were only observed for leukocytes, platelets, and neutrophils. Similar patterns of associations were observed by investigating the tertiles of each inflammatory maker (barplot in each panel). All P values were two-sided, and unadjusted. Size of the bar and the internal center represent mean HRs. The error bar and shadow indicate the corresponding 95% CI. **E** Eight out of the nine inflammatory markers significantly mediated the prospective association between physical frailty and depression incidence while adjusting for covariates and multiple comparisons. Path thickness indicates the strength of associations, and numerical values for the largest and smallest effect sizes are provided for reference. The number of participants with complete data for these inflammation markers were $n = 342,268$ (leukocyte), $n = 342,771$ (lymphocyte, neutrophil%, monocyte), $n = 342,775$ (lymphocyte%, monocyte%, neutrophil%), $n = 343,371$ (platelet), and $n = 352,277$ (CRP). Two-sided unadjusted P values and HRs were derived from Cox proportional hazard model (Z-tests). CRP, C-reactive protein. HR hazard ratio. Icons were made from https://www.svgrepo.com/. Source data are provided as a Source Data file.

depression after controlling for covariates and multiple comparisons (Bonferroni-corrected significance threshold $P < 0.05/9$, Fig. 3C, Supplementary Table 8). Specifically, neutrophils, leukocytes, and serum CRP showed comparably large effect sizes, where one SD increase corresponds to about 10% higher risk of developing depression. In comparison, the lymphocyte percentage was protective for depression, with one SD increase corresponding to a 5% lower risk.

There was evidence for non-linear associations of leukocytes ($P < 0.001$), platelets ($P = 2.04 \times 10^{-3}$), and neutrophils ($P = 4.42 \times 10^{-3}$) with increased depression incidence, with plateauing slopes at lower exposure and linear trends at higher levels (Bonferroni-corrected significance threshold $P < 0.05/9$, Fig. 3D). No evidence of non-linearity was observed for other inflammatory markers. Moreover, similar patterns of association were revealed by investigating the tertiles of each inflammatory marker (Supplementary Table 8).

Mediation analyses revealed a partial but significant mediation effect of eight out of the nine inflammatory markers on the prospective association between physical frailty and depression incidence independent of covariates (Fig. 3E). The proportion of mediated variance for significant mediations ranged from 0.17% (lymphocyte) to 1.84% (CRP levels).

## Replication of associations between frailty, inflammatory markers, and depression incidence in matched data

We used propensity score matching to generate two groups of 11,241 frail and non-frail participants matched on all 10 covariates (all $P > 0.05$, Supplementary Table 9). Analyses using the matched data yielded nearly unchanged results regarding the direction and magnitude of associations as those from the main analyses. Specifically, the risk of depression for frail individuals was 3.08-fold higher than the matched non-frail individuals (HR = 3.08, [2.74, 3.47], $P = 1.96 \times 10^{-77}$, Fig. 4A). Significant differences between frail and non-frail individuals were observed for all inflammatory markers other than lymphocytes (d = 0.03, P = 0.17), with serum CRP again showing the largest effect size (d = 0.48, $P < 0.001$, Fig. 4B, Supplementary Table 10). Moreover, based on the matched data, only CRP, neutrophils, and leukocytes significantly mediated the prospective association between frailty and depression incidence (Bonferroni-corrected two-side $P < 0.05/9$ for

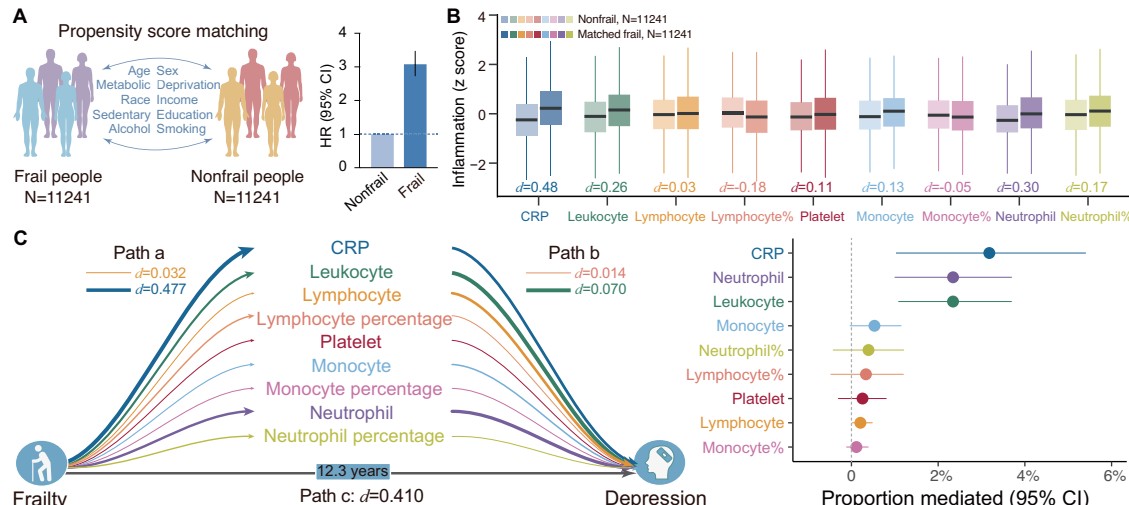

**Fig. 4 | Associations between physical frailty, inflammatory markers, and depression incidence in matched data. A** The propensity score matching procedure generated two groups of 11,241 frail and non-frail participants who were matched on all 10 covariates. The risk of depression for frail people was 3.08-fold higher than the matched non-frail people. The size of the bar represents mean HRs. The error bar represents the corresponding 95% CI. **B** Significant differences between frail and non-frail individuals were observed for all inflammatory markers other than lymphocytes (two-sided *T* test). Boxplot elements were defined as follows: the center line is the median value; box limits are the upper and lower quartiles; whiskers are 1.5× the interquartile range. **C** Of all inflammatory markers, only CRP, neutrophils, and leukocytes significantly mediated the prospective association between frailty and depression incidence (frail vs non-frail: *n* = 11,241 vs *n* = 11,241). The significance of mediating effects was determined based on 5000 bootstrap iterations. Path thickness indicates the strength of associations, and numerical values for the largest and smallest effect sizes are provided for reference. Dots: mean HRs; Horizontal lines: 95% CI. CRP C-reactive protein. HR hazard ratio. Icons were made from https://www.svgrepo.com/. Source data are provided as a Source Data file.

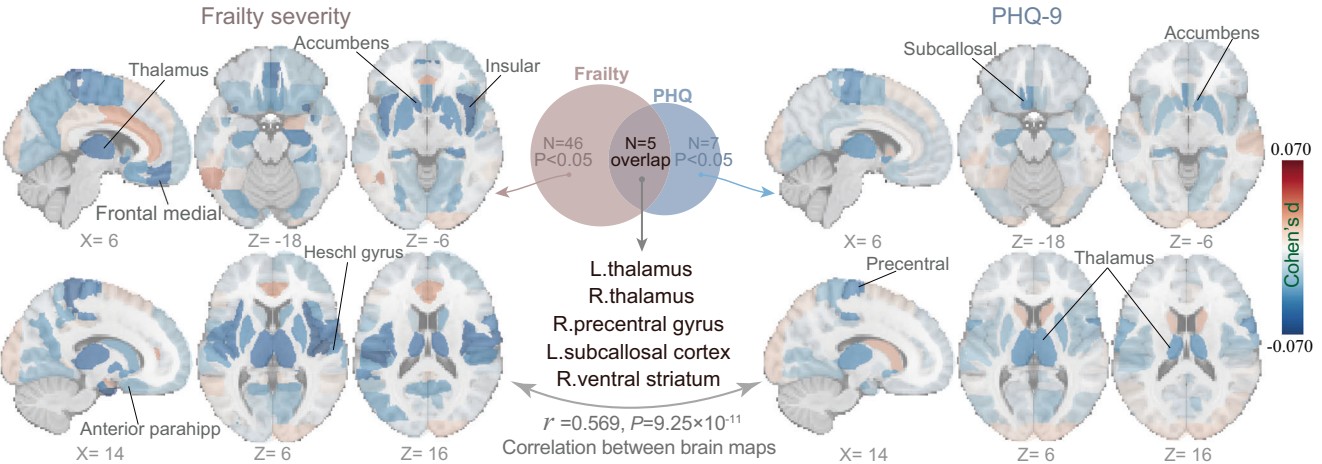

**Fig. 5 | Brain structure associated with physical frailty and symptoms of depression.** The neuroimaging analyses revealed 46 and 7 regional brain regions significantly associated with physical frailty and depression symptoms respectively, while adjusting for potential covariates and multiple comparisons. Five brain regions were consistently identified by both physical frailty and PHQ-9 scores. The association map of frailty was significantly similar to that of depression symptoms, indicating a shared neurobiological basis (r = 0.569, two-sided, unadjusted *P* = 9.25 × 10⁻¹¹, *t* test). PHQ: patient health questionnaire. Source data are provided as a Source Data file.

nine tests), with the proportion of mediated variance being 3.18% (95% CI [1.03%, 5.41%], P = 2.80 × 10⁻³), 2.34% ([1.00%, 3.69%], P = 8.00 × 10⁻⁴), and 2.34% ([1.09%, 3.69%], *P* = 2.00 × 10⁻⁴), respectively (Fig. 4C, Supplementary Table 11).

## Associations of physical frailty and depressive symptoms with regional GMV

A sample of 21,346 participants with complete data for regional GMV was retained for neuroimaging analyses. The neuroimaging analyses, after controlling for covariates and multiple comparisons, provided strong evidence for negative associations between physical frailty and 46 regional GMVs (*P* < 0.05, |d| = 0.03–0.07, Supplementary Table 12), as well as between PHQ-9 scores and seven regional GMVs (*P* < 0.05, |

d| = 0.038−0.048, Supplementary Table 13, Fig. 5). Notably, the association map of frailty was significantly similar to that of PHQ-9 scores (r = 0.569, *P* = 9.25 × 10⁻¹¹), and five brain regions showed consistent associations with both frailty and PHQ-9 scores, including left and right thalamus, right precentral gyrus, left subcallosal cortex, and right ventral striatum. Further analysis indicated that the mean GMV of these five brain regions significantly mediated the association between physical frailty and PHQ-9 scores (*P* < 0.001, proportion of mediated variance 0.72%, 95% CI = [0.33%, 1.00%], Supplementary Fig. 6).

## Discussion

This study demonstrates an increased risk of developing depression for individuals with pre-frailty and frailty during a 12-year period,

independent of many putative confounds. Moreover, several subcortical brain structures and three inflammatory markers—CRP, neutrophils, and leukocytes—significantly mediated the association between frailty and depression, suggesting that both neurobiological and inflammatory mechanisms underlie the frailty-depression relationship.

Existing studies on identifying modifiable risk factors for depression have been focused on lifestyle and environmental factors[3,5,20]. Our findings agree with previous literature showing disproportionate risks of depression in physical frailty[10,11,13,21]. The HR of depression in our study (HR$_{pre-frailty}$ = 1.6, HR$_{frailty}$ = 3.2) is similar to that observed in a meta-analysis with 8023 participants from 24 studies[11]. However, most studies included in this meta-analysis used smaller sample sizes and relied on a cross-sectional design, limiting inferences about prospective associations. Another study of 1827 Singapore adults identified that pre-frailty and frailty were associated with a 2.26- and 3.75-times increased risk of incident depression over a four-year follow-up[22]. Despite a prospective design, this study—alongside others—did not use the time-to-event analysis and determined depression based on symptom scales. Using symptom scales does not constitute a clinical diagnosis[5] and may be confounded by the overlap between self-reported depressive symptoms and signs of frailty[23–25]. Our work builds upon these studies and extends them in several critical aspects. The UK Biobank, which includes over 350,000 participants over a 12-year follow-up period, affords adequate statistical power, making results less likely to be generated by chance[26]. It also allows us to adequately adjust for covariates that are postulated to influence both frailty and depression.

Frailty research almost exclusively focuses on individuals aged 65 years and over[7,27]. However, our subgroup analyses indicated a greater risk of depression associated with frailty in middle-aged individuals than in late adulthood. Frail individuals younger than 65 years had a 3.37-fold increased risk of depression incidence compared with a 2.53-fold increased risk in those older than 65 years. This finding implies that frailty does not exclusively affect older people and can negatively impact mental health at a younger age. Previous work has ascribed the protective effect of age on depression to more effective coping mechanisms of older people in regulating emotional experiences[28,29]. Alternatively, health decline and frailty may be less expected in younger people, leading to a greater effect of frailty on well-being. Additional evidence suggests that frailty in later life can be reversed by adhering to healthy lifestyles in midlife[12], highlighting the importance of early assessment and intervention in middle age. Further, we observed greater risks in males than females. This result may be partly explained by age-related hormonal dysregulation, especially testosterone[30,31], but this merits further inquiry.

The contribution of physical frailty to depression risk is higher than that observed for all lifestyle and socioeconomic factors. In this context, screening for frailty may be a more effective tool for identifying individuals at high risk of depression. Indeed, physical frailty components are readily accessible and cost-effective when benchmarked against the disease burden of depression within primary care. Consequently, our study argues for incorporating frailty assessments as routine practice. Our study also contributes insights into emerging evidence that body health across distinct organ systems is essential to mental health[32–34]. Management of neuropsychiatric disorders, including depression, should acknowledge the importance of body health. Accordingly, components of physical frailty like weakness, walking speed, and physical activity are important parameters reflecting musculoskeletal health[35], while exhaustion and weight loss are measures of general physical health. Future studies identifying modifiable risk factors of depression can dissect phenotypes from other body systems and even integrate them to form a holistic measure to understand better how body health contributes to depression.

Our study provided additional evidence for mechanisms underlying the link between physical frailty and depression. Independent lines of research suggest elevated peripheral inflammation in depression[15,36,37] and frailty[18,38], indicating a role of immunoinflammatory dysregulation. We demonstrated similar associations in a prospective design and incorporated multiple inflammatory markers, including CRP, neutrophils, leukocytes, platelets, and monocytes. More importantly, our study showed significant mediating effects of CRP, neutrophils, and leukocytes on the frailty-depression association, suggesting that the effect of frailty on depression incidence may occur through inflammation-related processes[37]. Impaired homeostasis in frailty can stimulate the excess secretion of proinflammatory cytokines. These cytokines can traverse the blood-brain barrier and increase its permeability, leading to disordered synaptic plasticity and structural changes in the brain by activating microglia and astrocytes[39]. Proinflammatory cytokines can also lead to depressive behaviors by reducing serotonin, altering neurotransmitters' metabolism, and releasing corticotrophin hormone[40].

Our neuroimaging analyses revealed a widespread negative association of regional GMVs with physical frailty and depressive symptoms. More interestingly, the most consistently affected brain regions, including the thalamus, right precentral gyrus, left subcallosal cortex, and ventral striatum, also significantly mediated the association between frailty and depressive symptoms, lending support to the hypothesis that physical frailty and depression share an overlapping neurobiological basis. These brain regions also show the most rapid age-related losses, suggesting that accelerated brain aging may be the underlying mechanism linking physical frailty with depression[41,42].

We detected relatively small mediation effects, consistent with other large-scale studies[7,43]. One potential explanation is our adequate adjustment of covariates that co-occur with frailty and depression. The mediation effects should be interpreted as the unique contribution of inflammatory markers or brain structure beyond confounds. The sickness behavior theory posits that inflammatory markers can only initiate neurovegetative symptoms in a subset of individuals[16]. Likewise, inflammation and brain structure may only account for part of the underlying mechanisms. Other implicated mechanisms include insulin resistance, oxidative stress, gut microbiota, and genetic susceptibility[11,21,44]. Nevertheless, the effect sizes of inflammation and brain structure are biologically meaningful and have significance for establishing guidelines in a large population.

The present finding agrees with our previous study showing the association between physical frailty and depression[7], but is also distinct from the previous work in notable ways. First, in the previous study, we used a data-driven approach to investigate how physical frailty relates to more than 300 phenotypes, among which depression-related measures exhibited the strongest relationships. The current study was hypothesis-driven, examining the prospective relationship between physical frailty and depression incidence. Second, the current study was based on longitudinal data and used time-to-event analysis to examine the prospective association between baseline frailty and follow-up depression incidence. In comparison, the previous study was primarily based on cross-sectional data and used linear mixed-effect models to examine associations. Third, the current study provided additional evidence for the mediating effects of inflammatory markers and the causal relationships between frailty and depression, which was absent from previous studies.

Some limitations need to be acknowledged. First, significant mediation effects are measures of association[45] and do not allow disentangling the causality. Furthermore, although the causal inferences of physical frailty on depression was demonstrated using MR, these estimates should be interpreted with caution until further confirmation in randomized controlled trials[5]. Estimates from MR analysis reflect lifelong average effects of genetic variants on depression. Second, reporting bias exists as four of the five frailty indicators were self-reported. However, the self-reported data also bears significant strengths. They are less time-consuming to collect and are feasible in

routine primary care practice[7]. Further, the self-reported frailty phenotype is comparable with objectively measured alternatives in predicting adverse health outcomes[46]. Third, participants in the UK Biobank cohort are predominantly of European ancestry, tend to be less deprived, and have more healthy behaviors than the wider UK population, which necessitates significant replication in diverse populations. Fourth, although we focused on GMVs in the neuroimaging analyses due to their broad significance in brain atrophy, future research can explore other imaging measures. In addition, the current study examined the mediating effect of nine inflammatory markers. Inflammatory markers reflecting more specific depression-related pathways warrant further inquiry[47]. Fifth, perhaps counterintuitive, our finding indicated that frail and pre-frail participants drank alcohol less frequently than non-frail individuals. This result can be explained by abstainer bias, where the non-drinking group often includes former drinkers who have given up alcohol because of health problems[48,49]. Our additional analyses suggest that former drinkers had a significantly increased risk for depression than those who never drink or current drinkers (Supplementary Table 14). Sixth, unmeasured or residual confounding is inevitable. However, notwithstanding these limitations, the prospective design, large sample sizes, extended follow-up, consistency of results in sensitivity analyses, and replication of findings on matched data partly allay concerns over reverse causality of the observed associations and reassure that our results are valid.

This study demonstrated increased risks of incident depression in pre-frail and frail individuals and provided insights into the neurobiological and inflammatory mechanisms linking physical frailty to depression. Future work should determine whether frailty is amenable to intervention for depression. If so, routine surveillance and assessment of physical frailty could be integrated into current clinical practices to help determine depression risk. This approach might be especially true for middle-aged adults, who appear to have a higher association between frailty and depression risk than in late adulthood.

## Methods

### Study population
The UK Biobank is a population-based cohort study of over 500,000 participants aged 37–73 years[50]. The UK Biobank study was approved by the North West Multicenter Research Ethics Committee (No.11/NW/0382), and written informed consent was obtained from all participants. This research was conducted using the UK Biobank resources (application number 42009). Between 2006 and 2010, participants attended one of 22 assessment centers, where they completed touchscreen and nurse-led questionnaires, had physical measurements taken, and provided biological samples. Since 2014, a subsample of participants was invited back for imaging assessment. We excluded participants who had missing data or responded "prefer not to answer" or "do not know" for any covariates and five frailty indicators. Details for participant selection are provided in Supplementary Fig. 7.

### Assessment of physical frailty
We adopted the physical frailty phenotype to assess frailty severity[19], which included the following five indicators: weight loss, exhaustion, weakness, physical inactivity, and slow walking speed. As per previous studies[7,14,51], some items were adapted to accommodate data available within UK Biobank. Details can be found in Supplementary Table 15. Notably, weakness (assessed by handgrip strength) was measured using a hydraulic hand dynamometer, while the other four indicators were self-reported. Responses for the five components were coded as yes or no, signifying whether the criterion was met. Then, the number of criteria met was summed to indicate the severity of frailty, resulting in a score ranging from 0 to 5. For consistency with the literature[14,52,53], three mutually exclusive groups of frailty, pre-frailty, and non-frailty were defined for participants fulfilling three or more, one or two, or no criteria, respectively.

### Ascertainment of depression
Depression diagnosis was defined as ICD-10 codes F32 (depressive episode) or F33 (recurrent depressive disorder) of 'the first occurrence data' fields generated by UK Biobank, which is identified through linkage to inpatient hospital data, death register, self-report (based on physical diagnosis and subsequently confirmed with nurses), and primary care[2]. Individuals with preexisting depression at baseline were excluded from our sample. We also performed a 2-year landmark analysis to minimize possible reverse causality by excluding participants who experienced events within the first 2 years of follow-up. We calculated the follow-up time from the baseline to the time of depression diagnosis, death, loss to follow-up, or the censoring date, whichever occurred first.

Symptoms of depression were measured at the imaging assessment using the 9-item version of the Patient Health Questionnaire (PHQ-9) rather than clinical depression diagnosis due to a limited number of participants diagnosed with depression when the imaging was performed (about 2.60%, see Supplementary Methods Page 24).

### Assessment of inflammatory markers and brain imaging data
Based on their sensitivity to different inflammatory processes and the availability in the UK Biobank, nine inflammatory markers were included in this study, which were extracted from baseline blood samples. Blood cell counts were performed using a Beckman Coulter LH750 to determine the total number of leukocytes and platelets, plus neutrophils, lymphocytes, and monocytes (as well as their percentage in leukocytes). Serum C-reactive protein (CRP) was measured by immunoturbidimetric high-sensitivity analysis on a Beckman Coulter AU5800. Details regarding the blood sample processing can be found at https://biobank.ndph.ox.ac.uk/showcase/label.cgi?id=100080. Data with significantly skewed distributions were log-transformed and standardized to z-scores prior to analyses.

Brain MRI data were acquired and processed by the UK Biobank team and made available to approved researchers as image-derived phenotypes (IDP). IDPs used in this study included regional gray matter volumes (GMV) of 96 cortical and 14 subcortical regions, which were normalized for head size[54]. Details regarding data acquisition protocols and preprocessing can be found at https://biobank.ctsu.ox.ac.uk/crystal/crystal/docs/brain_mri.pdf and elsewhere[55,56]. Field identifications used in this study are described in Supplementary Table 16.

### Covariates
A total of 10 covariates were included in all analyses. Age was determined from the date of birth at baseline assessment and dichotomized as middle-aged and older groups using 65 years as the cutoff. Sex was self-reported. Race was self-reported and dichotomized as white and non-white. Area-based socioeconomic status was measured using the Townsend score and divided into tertiles. Educational attainment was self-reported and dichotomized as with and without university or college degree-level qualifications. Average total household income was self-reported and categorized as low, middle, and high. Self-reported smoking status was categorized as never and ever smokers. Participants reporting television watching time over four hours each day were classified as having sedentary behavior. Self-reported alcohol intake frequency ranged from 'never' to 'daily or almost daily' and was coded as a categorical variable. Metabolic syndrome was defined as the occurrence of any three or more of the following components: central obesity, diabetes, hypertension, low HDL, and high triglycerides. Details regarding the measurements of covariates are described in Supplementary Methods (Page 23). These covariates were chosen on the basis of the existing literature[2,4,5,20,57,58] investigating the association between modifiable risk factors and incident depression using UK Biobank data. All included covariates have individually been implicated in depression.

## Prospective association between physical frailty and depression incidence

Cox proportional hazard models were used to estimate the association between frailty status and depression incidence with follow-up time as the time-dependent variable. Individuals classified as non-frail were used as the reference group, and hazard ratios (HR) and 95% CI were calculated for pre-frail and frail individuals. To evaluate whether the depression risk increased along with the number of frailty indicators, we also established six-category models by treating frailty scores as a categorical variable and using individuals fulfilling no frailty indicator as the reference group. The non-linear effects were examined by introducing a restricted cubic spline into the Cox proportional hazard models. Age, sex, race, alcohol intake, smoking status, sedentary behavior, education level, material deprivation, family income, and metabolic syndrome were treated as covariates being associated with both exposures and outcomes in all models. The proportional hazard assumption was checked using Schoenfeld residuals; no violations were observed. Models showed acceptably low multicollinearity (variance inflation factor<1.5).

To assess the modifying effect of covariates, we included a multiplicative interaction term for each of the 10 covariates and ran further subgroup analyses by covariates showing significant interactions. We also examined the association between five frailty components and depression risks individually (by including only one component of frailty as an exposure variable) and mutually (by including all five components simultaneously)[52]. Furthermore, the population attributable fraction[26,59] was calculated to estimate the proportion of depression cases attributable to both frailty and its components.

## Association of inflammatory markers and brain structure with physical frailty and depression

Linear-mixed effect models were applied to investigate the association of physical frailty and depression symptoms with inflammatory markers and regional GMVs[60–63]. Within the same analytical framework, physical frailty (or PHQ-9) was fitted as a fixed effect, UK Biobank assessment center as a random effect, and each of the nine inflammatory markers (or regional GMVs) was set as the dependent variable in separate models. The same set of covariates as listed above were used here. We extracted the standardized beta coefficients and converted them to Cohen's $d$ according to a previous study[64].

Linear and nonlinear associations between inflammatory markers and depression incidence were also examined using Cox proportional hazards models within the same analytical framework. We first treated them as continuous variables and then constructed models by tertile for each inflammatory marker, using participants in the lowest tertile as the reference group. All analyses were corrected for multiple comparisons using Bonferroni correction. Brain association analyses were corrected using the Benjamini-Hochberg false discovery rate (FDR) method due to the many brain regions.

## Mendelian randomization analyses

We performed two-sample MR[65] analyses for causal inference between genetically predicted physical frailty and the risk of depression using the TwoSampleMR package in R[66]. UK Biobank-based summary statistics for physical frailty were obtained from a recent study[67] based on 386,565 participants of European descent. We extracted a total of 30 highly associated SNPs ($P < 5 \times 10^{-8}$) that were clumped for independence at $r^2 > 0.001$ with a window of 10,000 kb based on European ancestry reference data from the 1000 Genomes Project. These SNPs were used as instrument variables. The summary statistics for depression were obtained from large consortia with European samples from Psychiatric Genomics Consortium that left out UK Biobank data to minimize sample overlap, which included a total of 135,458 cases and 344,901 controls[68]. Samples from 23andMe were also left out owing to general access constraints. The IVW method was

implemented as the primary method. Since no single method exists that outperforms all others[69], the weighted median and MR-Egger were implemented as sensitivity methods. The heterogeneity induced by different genetic variants in the fixed-effect IVW method was assessed by Cochran's Q test. If significant heterogeneity exists, the effect would be estimated using the IVW method under multiplicative random effect[13]. The pleiotropy was quantified with the MR-Egger regression intercept term. Leave-one-SNP-out analysis was performed to assess if the overall effect was driven by any single SNP. MR-PRESSO was employed to detect outliers and we removed these outliers to generate reported estimates[70]. For sensitivity analyses, we extracted highly associated SNPs under a relaxed threshold ($P < 5 \times 10^{-7}$) and reran the above MR analysis.

## Mediation analyses

Using the "mediation" package in R, we established a standard three-variable path model to investigate how much of the associations between physical frailty and depression incidence can be mediated by brain structure or inflammatory markers[71,72]. Linear regression models were used for frailty-inflammatory associations, and survival regression was used for inflammation-depression and frailty-depression models[72]. In examining the mediating effect of brain structure, mean GMVs of brain regions were modeled as the mediator, and PHQ-9 scores as the dependent variable. The same set of covariates used in the association analyses was included here. The significance of mediating effects was determined based on 5000 bootstrap iterations.

## Replication based on matched samples

To validate the results[73], we conducted a propensity score matching analysis using the "MatchIt" package in R[74]. Participants with frailty were matched on all 10 covariates to a single participant with non-frailty (1:1 ratio nearest-neighbor matching without replacement). Based on the matched data, we investigated the associations between frailty status and depression incidence using Cox proportional hazards models and also examined the mediating effect of inflammatory markers.

## Reporting summary

Further information on research design is available in the Nature Portfolio Reporting Summary linked to this article.

## Data availability

The UK Biobank data are available via their standard data access procedure at https://www.ukbiobank.ac.uk/ with access fees. Researchers can apply for access to the UK Biobank data via the Access Management System (AMS) (https://www.ukbiobank.ac.uk/enable-your-research/apply-for-access). This research was conducted using the UK Biobank resources (application number 42009). Summary statistics from previous GWAS study of physical frailty are publicly available on Figshare https://figshare.com/s/6683396c68807fe4e729. Summary statistics for depression were obtained from Psychiatric Genomics Consortium and can be downloaded from https://figshare.com/articles/dataset/MDD2_MDD2018_GWAS_sumstats_w_o_UKBB/21655784. Source data are provided with this paper.

## Code availability

Scripts used to perform the analyses are available at https://github.com/Jiang-brain/frailty-depression[75].

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

## Acknowledgements

All data used in this study are publicly accessible from UK Biobank via their standard data access procedure (https://www.ukbiobank.ac.uk/). We are grateful to all participants who donated their time to this project and the UK Biobank team for collecting, processing, and disseminating data used in this study. This work is supported in part by the National Institute of Mental Health (NIMH, R01MH118695, V.D.C.; R01MH117107, V.D.C.; R00MH130894, S.N.), National Science Foundation (2112455, V.D.C.), the National Natural Science Foundation of China (62373062, 82022035, J.S.), the National Science Foundation Graduate Research Fellowship (DGE2139841, M.R.).

## Author contributions

R.J., J.S., and D.S. conceptualized the study. R.J. and W.D. curated the data. R.J. performed the formal analyses and drafted the manuscript. R.J., W.D., and S.Q. contributed to the visualization. V.D.C., S.L., J.Y., M.R., S.N., and J.S. reviewed and edited the manuscript. All authors read and approved the final manuscript.

## Competing interests

The authors declare no competing interests.

## Additional information

**Peer review information** : *Nature Communications* thanks Klaus Ebmeier, and the other, anonymous, reviewer(s) for their contribution to the peer review of this work. A peer review file is available.

