## [Peer Review File · Nature Communications]

The brain structure, inflammatory, and genetic mechanisms mediate the association between physical frailty and depressionREVIEWER COMMENTS

Reviewer #1 (Remarks to the Author):

This is a longitudinal study of the association of initial frailty and subsequent depression in a 12-year follow-up study. Initial frailty predicted depression at follow-up. Frailty was associated with inflammatory markers at baseline. Inflammatory markers predicted depression at follow-up. Frailty effects on depression at follow-up were partially mediated by some inflammatory markers. These effects persisted in part after controlling for 10 putative covariates by propensity score matching. There was a "negative association between physical frailty and 46 regional GMVs" like that found with (PHQ-9) depression scores. Support of a causal association by a Mendelian Randomisation analysis is claimed and the briefest summary of the analysis is given in the supplement, but not enough information is supplied to verify this claim. In summary, the strong claims for a new approach to prevent depression are not warranted by the data supplied.

Please, also explain in detail, how the data in this paper differ from and overlap with those presented in your paper in *Lancet Digit Health* 2023; 5: e350–59.

Abstract: "routine surveillance of physical frailty from middle age may provide cost-effective targets for potentially mitigating the onset of depression" – you are both hedging and contracting your claim here: is it possible to be more transparent about what you are trying to achieve? Do you mean something like 'for identifying potential modifiable causes of depression' that may then hopefully be amenable to intervention? Your formulation seems to imply that reducing frailty or normalising C-reactive protein levels, neutrophil or leukocyte numbers (or even expanding certain brain volumes?) are effective prophylactic measures that can prevent depression onset. Unfortunately, the 'translational path' is much longer.

Introduction: "Specifically, evidence suggests that about 14% of depression cases could be prevented by adhering to favourable lifestyles, including a healthy diet, appropriate weight, and non-smoking [6]." - This refers to a paper describing associations which are liable to residual confounding, so deducing a potential for prevention is not warranted by the data.

"However, several critical knowledge gaps remain." - What about an absence of studies showing a causal connection between frailty and depression? Or put differently, the possibility of residual confounding in the available studies? Please only supply either peer-reviewed results or the full method and analysis of extant studies.

Reviewer #1 (Remarks on code availability):

Reviewer #2 (Remarks to the Author):

Jiang et al. conducted a comprehensive investigation into the relationship between physical frailty and depression, considering brain structure, inflammatory factors, and genetic aspects. I was impressed by the thorough analysis carried out in this study. In my opinion, the paper adds new value and insights into the underlying mechanisms between these two traits. My comments are as follows:

1. I suggest that the authors include an overall flowchart of the entire study. While I noticed a flowchart in Figure S1 highlighting the sample selection, it would be highly beneficial to have a flowchart illustrating the main analyses conducted in this study.
2. Could the authors explain the rationale behind selecting the 10 covariates?
3. In the results section, the authors found that "frail and pre-frail participants are more likely to smoke and spend more time in sedentary behaviors but drink alcohol less frequently than non-frail individuals." This is an intriguing result, suggesting that alcohol consumption may protect individuals from frailty. Could the authors provide further explanation and discussion on this finding?
4. In the results section, the authors used PHQ-9 scores to assess the association between

depressive symptoms and regional GMV. Could the authors clarify why they did not divide the sample into depression cases and controls for the association analysis?

5. Could the authors explain the rationale behind selecting the 9 inflammatory markers?

I recommend the publication of this paper once the authors address the above concerns.

Reviewer #2 (Remarks on code availability):

The code provide a README file with enough instructions for installing and running the applicatio

We would like to thank the editor and reviewers for providing valuable feedback about our manuscript and giving us further opportunities to revise it. We considered each comment carefully and made extensive revisions. We are confident that all questions have been addressed and believe that the current version is significantly improved.

This response includes each of the reviewers' comments with our responses (in blue). The corresponding updates are also provided (in blue) following the response to each comment. For the editor's and reviewers' convenience, we have shown all updates in the manuscript and Supplement with colour highlighting (in blue).

Reviewer #1 (Remarks to the Author):

This is a longitudinal study of the association of initial frailty and subsequent depression in a 12-year follow-up study. Initial frailty predicted depression at follow-up. Frailty was associated with inflammatory markers at baseline. Inflammatory markers predicted depression at follow-up. Frailty effects on depression at follow-up were partially mediated by some inflammatory markers. These effects persisted in part after controlling for 10 putative covariates by propensity score matching. There was a "negative association between physical frailty and 46 regional GMVs" like that found with (PHQ-9) depression scores.

We thank the reviewer for the positive and insightful comments. To improve the manuscript, we have made extensive revisions as detailed below.

1. Support of a causal association by a Mendelian Randomisation analysis is claimed and the briefest summary of the analysis is given in the supplement, but not enough information is supplied to verify this claim. In summary, the strong claims for a new approach to prevent depression are not warranted by the data supplied.

We agree with the reviewer that more details are needed to support the causal inferences. In response to the reviewer, we provided more details on the Mendelian randomization analysis and moved it from Supplement to the main text. Moreover, we performed the following analyses to confirm the robustness of the MR estimates including:

- We used Cochran's Q test to examine the heterogeneity induced by different genetic variants in the fixed-effect IVW method. Due to the existence of significant heterogeneity, the IVW method under multiplicative random effect was used as the primary method¹ in the manuscript.
- We performed a leave-one-SNP-out analysis to assess if the overall effect was driven by any single SNP.

- The MR-PRESSO outlier test was employed to detect outliers and we removed these outliers to generate reported estimates².
- For sensitivity analyses, we extracted highly associated SNPs under a relaxed threshold ($P < 5 \times 10^{-7}$) and reran the above MR analysis.

We added a new section ‘Causal inferences of the effect of physical frailty on depression’ in the Results as follows:

“We performed Mendelian randomization (MR) analysis for causal inference between genetically predicted physical frailty and the risk of depression. The MR-Egger intercept term indicated no obvious directional pleiotropy (intercept=0.003, $P=0.810$), but Cochran’s Q test revealed significant heterogeneity ($Q=79.89$, $P=1.20 \times 10^{-6}$). Thus, the inverse-variance weighted (IVW) method under multiplicative random effect was used as the primary method¹. Using 30 significantly frailty-associated SNPs (single nucleotide polymorphisms, **Table S6**) as proxies, the IVW method found that each one-point increment in physical frailty was associated with a 2.55-times higher risk of depression (odds ratio (OR) =2.55, 95% CI= [1.59, 4.08], $P=1.01 \times 10^{-4}$, **Figure S3**). The MR estimates based on the weighted median (OR =1.92, 95% CI= [1.19, 3.10], $P=7.96 \times 10^{-3}$) and MR-Egger (OR =2.02, 95% CI= [0.29, 13.95], $P=0.482$) showed consistent detrimental effects of physical frailty on depression, but was only significant for the weighted median. MR-PRESSO detected three outliers, the removal of which had a nominal impact on effect estimates (OR =2.73, 95% CI= [1.82, 4.10], $P=1.37 \times 10^{-6}$). Further, leave-one-SNP-out analyses suggest that no single SNP drove these MR estimates (**Figure S3**). Sensitivity analysis under a relaxed threshold for selecting significant instruments ($P < 5 \times 10^{-7}$) yielded similar results as the primary finding (**Figure S4**), providing further evidence for the stability of the causal inferences.”

The description of Mendelian randomization has been moved from Supplement to the Methods section and updated as follows:

“We performed two-sample MR³ analyses for causal inference between genetically predicted physical frailty and the risk of depression using the TwoSampleMR package in R⁴. UK Biobank-based summary statistics for physical frailty were obtained from a recent study⁵ based on 386,565 participants of European descent. We extracted a total of 30 highly associated SNPs ($P < 5 \times 10^{-8}$) that were clumped for independence at $r^2 > 0.001$ with a window of 10000 kb based on European ancestry reference data from the 1000 Genomes Project. These SNPs were used as instrument variables. The summary statistics for depression were obtained from large consortia with European samples from Psychiatric Genomics Consortium that left out UK Biobank data to minimize sample overlap, which included a total of 135,458 cases and 344,901 controls⁶. Samples from

23andMe were also left out owing to general access constraints. The IVW method was implemented as the primary method. Since no single method exists that outperforms all others⁷, the weighted median and MR-Egger were implemented as sensitivity methods. The heterogeneity induced by different genetic variants in the fixed-effect IVW method was assessed by Cochran's Q test. If significant heterogeneity exists, the effect would be estimated using the IVW method under multiplicative random effect¹. The pleiotropy was quantified with the MR-Egger regression intercept term. Leave-one-SNP-out analysis was performed to assess if the overall effect was driven by any single SNP. MR-PRESSO was employed to detect outliers and we removed these outliers to generate reported estimates². For sensitivity analyses, we extracted highly associated SNPs under a relaxed threshold ($P < 5 \times 10^{-7}$) and reran the above MR analysis."

Figure S3. Mendelian randomization plots for physical frailty \rightarrow risk of depression with SNPs selected under the threshold of 5.0×10^{-8} . (A) The causal inferences of the effect of physical frailty on depression based on Mendelian randomization analyses. (B) Scatterplot of SNP effects on physical frailty versus their effects on depression, with the

slope of each line representing estimated MR effect per method. (C) Forest plot of leave-one-SNP-out sensitivity analysis. The significant SNPs were selected under the threshold of 5.0×10^{-8} , and three outliers detected by MR-PRESSO were removed from the MR analysis.

Figure S4. Mendelian randomization plots for physical frailty \rightarrow risk of depression with SNPs selected under the threshold of 5.0×10^{-7} . (A) The causal inferences of the effect of physical frailty on depression based on Mendelian randomization analyses. (B) Scatterplot of SNP effects on physical frailty versus their effects on depression, with the slope of each line representing estimated MR effect per method. (C) Forest plot of leave-one-SNP-out sensitivity analysis. The significant SNPs were selected under a threshold of 5.0×10^{-7} , and one outlier detected by MR-PRESSO was removed from the MR analysis.

We also acknowledge the interpretation of MR estimates as a potential limitation as follows:

Page 13: “Furthermore, although the causal inferences of the effect of physical frailty on depression was demonstrated using MR, these estimates should be interpreted with

caution until further confirmation in randomized controlled trials⁸. Estimates from MR analysis reflect lifelong average effects of genetic variants on depression.”

2. Please, also explain in detail, how the data in this paper differ from and overlap with those presented in your paper in *Lancet Digit Health* 2023; 5: e350–59.

We thank the reviewer for bringing this important issue to our attention. Both studies used data from the UK Biobank and consistently found that physical frailty is associated with depression. However, the current study is distinct from the previous work in notable ways including:

- The work in *Lancet Digital Health* was data-driven. In it, we investigated how physical frailty relates to more than 300 health-related outcomes and observed the strongest effects on depression-related measures. The current study was hypothesis-driven, examining the prospective relationship between physical frailty and depression incidence on the basis of previous work showing their cross-sectional associations. The work published in *Lancet Digital Health* is a motivation for the present study.
- The previous work was primarily based on cross-sectional data and used linear mixed-effect models to examine the association between physical frailty and depressive symptoms assessed by PHQ-9 scores. In comparison, the present study was based on longitudinal data and used the time-to-event analysis to examine the prospective association between baseline frailty and follow-up depression incidence. The diagnosis of depression was identified through linkage to inpatient hospital data, death register, self-report (based on physical diagnosis and subsequently confirmed with nurses), and primary care.
- Apart from showing the association between physical frailty and incident depression, the present study also investigated the mediating effects of inflammatory markers, providing additional evidence for the potential mechanisms.
- The present study provided further evidence for the causal relationship between physical frailty and depression using Mendelian Randomization analysis, which is absent from previous studies.

We have added the above discussion to the main text as follows:

Page 13: “The present finding agrees with our previous study showing the association between physical frailty and depression⁹, but is also distinct from the previous work in notable ways. First, in the previous study, we used a data-driven approach to investigate how physical frailty relates to more than 300 phenotypes, among which depression-related measures exhibited the strongest relationships. The current study was

hypothesis-driven, examining the prospective relationship between physical frailty and depression incidence. Second, the current study was based on longitudinal data and used time-to-event analysis to examine the prospective association between baseline frailty and follow-up depression incidence. In comparison, the previous study was primarily based on cross-sectional data and used linear mixed-effect model to examine associations. Third, the current study provided additional evidence for the mediating effects of inflammatory markers and the causal relationships between frailty and depression, which was absent from previous studies.”

3. Abstract: “routine surveillance of physical frailty from middle age may provide cost-effective targets for potentially mitigating the onset of depression “– you are both hedging and contracting your claim here: is it possible to be more transparent about what you are trying to achieve? Do you mean something like ‘for identifying potential modifiable causes of depression’ that may then hopefully be amenable to intervention? Your formulation seems to imply that reducing frailty or normalising C-reactive protein levels, neutrophil or leukocyte numbers (or even expanding certain brain volumes?) are effective prophylactic measures that can prevent depression onset. Unfortunately, the ‘translational path’ is much longer.

We apologize for the lack of clarity. We have updated the relevant sentences as follows:

Abstract: “Given the scarcity of curative treatment for depression and the high disease burden, identifying potential modifiable risk factors of depression, such as frailty, is needed.”

Conclusion: “...Identifying frailty as a modifiable risk factor is a necessary first step. Future work should determine whether frailty is amenable to intervention for depression. If so, routine surveillance and assessment of physical frailty could be integrated into current clinical practices to help determine depression risk. This approach might be especially true for middle-aged adults, who appear to have a higher association between frailty and depression risk than in late adulthood.”

4. Introduction: "Specifically, evidence suggests that about 14% of depression cases could be prevented by adhering to favourable lifestyles, including a healthy diet, appropriate weight, and non-smoking [6]." - This refers to a paper describing associations which are liable to residual confounding, so deducing a potential for prevention is not warranted by the data.

We thank the reviewer for this suggestion. We have now removed the above description and cited studies with more solid data (Choi *et al*, Am J Psychiatry, 2020; Choi *et al*, JAMA Psychiatry, 2019; Zhao *et al*, Nature Mental Health, 2024) as follows:

Page 3: “Many modifiable factors have been explored as components of prevention strategies including increased physical activity, decreased daytime napping, and reduced sedentary behaviors^{8, 10, 11}.”

5. "However, several critical knowledge gaps remain." - What about an absence of studies showing a causal connection between frailty and depression? Or put differently, the possibility of residual confounding in the available studies? Please only supply either peer-reviewed results or the full method and analysis of extant studies.

We apologize for the lack of clarity. We have acknowledged the absence of the causal relationship between physical frailty and depression as a knowledge gap in the Introduction section as follows:

Page 3: “Second, previous studies did not adequately account for many confounds occurring with frailty and depression, including metabolic syndrome and unhealthy lifestyles, which may obscure the true associations. Furthermore, observational studies cannot draw causal inferences due to residual confounding and potential reverse causality. Only one recent study has examined the causal relationship between frailty and depression¹, but they assessed frailty using the health deficit accumulation approach, which is less modifiable than the frailty phenotype¹².”

References

1. Deng M-G, Liu F, Liang Y, *et al*. Association between frailty and depression: A bidirectional Mendelian randomization study. *Science Advances* **9**, eadi3902 (2023).
2. Hemani G, Bowden J, Davey Smith G. Evaluating the potential role of pleiotropy in Mendelian randomization studies. *Hum Mol Genet* **27**, R195-R208 (2018).
3. Davies NM, Holmes MV, Davey Smith G. Reading Mendelian randomisation studies: a guide, glossary, and checklist for clinicians. *BMJ* **362**, k601 (2018).
4. Hemani G, Zheng J, Elsworth B, *et al*. The MR-Base platform supports systematic causal inference across the human phenome. *eLife* **7**, (2018).
5. Ye Y, Noche RB, Szejko N, *et al*. A genome-wide association study of frailty identifies significant genetic correlation with neuropsychiatric, cardiovascular, and inflammation pathways. *Geroscience* **45**, 2511-2523 (2023).

6. Wray NR, Ripke S, Mattheisen M, *et al.* Genome-wide association analyses identify 44 risk variants and refine the genetic architecture of major depression. *Nat Genet* **50**, 668-681 (2018).
7. Hypponen E, Mulugeta A, Zhou A, Santhanakrishnan VK. A data-driven approach for studying the role of body mass in multiple diseases: a phenome-wide registry-based case-control study in the UK Biobank. *Lancet Digit Health* **1**, e116-e126 (2019).
8. Choi KW, Stein MB, Nishimi KM, *et al.* An Exposure-Wide and Mendelian Randomization Approach to Identifying Modifiable Factors for the Prevention of Depression. *The American journal of psychiatry* **177**, 944-954 (2020).
9. Jiang R, Noble S, Sui J, *et al.* Associations of physical frailty with health outcomes and brain structure in 483 033 middle-aged and older adults: a population-based study from the UK Biobank. *Lancet Digit Health* **5**, e350-e359 (2023).
10. Zhao Y, Yang L, Sahakian BJ, *et al.* The brain structure, immunometabolic and genetic mechanisms underlying the association between lifestyle and depression. *Nature Mental Health* **1**, 736-750 (2023).
11. Choi KW, Chen C-Y, Stein MB, *et al.* Assessment of bidirectional relationships between physical activity and depression among adults: a 2-sample Mendelian randomization study. *JAMA psychiatry* **76**, 399-408 (2019).
12. Hoogendijk EO, Afilalo J, Ensrud KE, *et al.* Frailty: implications for clinical practice and public health. *Lancet* **394**, 1365-1375 (2019).

Reviewer #2 (Remarks to the Author):

Jiang et al. conducted a comprehensive investigation into the relationship between physical frailty and depression, considering brain structure, inflammatory factors, and genetic aspects. I was impressed by the thorough analysis carried out in this study. In my opinion, the paper adds new value and insights into the underlying mechanisms between these two traits. My comments are as follows:

We are grateful for the reviewer's positive comments.

1. I suggest that the authors include an overall flowchart of the entire study. While I noticed a flowchart in Figure S1 highlighting the sample selection, it would be highly beneficial to have a flowchart illustrating the main analyses conducted in this study.

We thank the reviewer for this insightful suggestion. For clarity, we have now added a new Figure 1 to show the study flowchart as follows:

Figure 1. Overview of analyses performed in the current study. Leveraging data from the UK Biobank, this study aims to investigate the association and causal inference between physical frailty and incident depression, and the potential mechanisms driven by inflammatory markers and brain structure. These analyses were replicated based on matched frail and non-frail samples to validate the main results.

2. Could the authors explain the rationale behind selecting the 10 covariates?

We apologize for the lack of clarity. These covariates were chosen on the basis of the existing literature^{1, 2, 3, 4, 5, 6} investigating the association between modifiable risk factors and incident depression using UK Biobank data. All included covariates have individually been implicated in depression and constitute the most common variables that

would be adjusted in epidemiological studies. Some covariates are less commonly adjusted in existing studies using UK Biobank but also show significant associations with depression. For completeness, we included another five factors into the current covariate set and reran the Cox proportional hazard models. The newly added covariates included:

- Current employment status (Field ID: 6142): a binary variable indicating whether individuals positively endorsed paid or self-employment.
- Air pollution level (Field ID: 24006): annual mean air pollution concentrations of particulate matter air pollution (pm2.5), which were coded as tertiles.
- Household size (Field ID: 709): derived from the question: ‘Including yourself, how many people are living together in your household?’ For this analysis, the answers were combined into three categories: single-occupancy households, households of two individuals and households of three or more⁷.
- Sleep duration (Field ID: 1160): derived from self-reported sleep duration and were categorized as optimal sleep (7-9 hours), deficient sleep (<7 hours), and excessive sleep (>9 hours).
- Cancer history (Field ID: 2453): derived from self-report and categorized as yes or no.

Results suggest that additionally controlling for these covariates had minimal influence on the results ($HR_{\text{prefrailty}} = 1.57$, 95% CI=[1.51, 1.64], $P=2.29 \times 10^{-101}$; $HR_{\text{frailty}} = 2.98$, 95% CI=[2.77, 3.21], $P=2.53 \times 10^{-185}$). We have added the above analyses to a new Supplement Table S3, and clarified it in the main text as follows:

Page 16: “These covariates were chosen on the basis of the existing literature^{1, 2, 3, 4, 5, 6} investigating the association between modifiable risk factors and incident depression using UK Biobank data. All the included covariates have individually been implicated in depression.”

Page 6: “These results did not change appreciably when using a 10-year landmark analysis or including an extended set of covariates (**Figure S2, Table S**)”

Table S3. Associations between frailty status and incident depression when including an extended set of covariates including employment status, air pollution, household size, sleep duration, and cancer history

	Participants	Events	HR (95% CI)	P value
Frailty status				
Non-frail	182772	4234	1 (Ref)	
Prefrail	126212	5255	1.57 (1.51-1.64)	2.29×10^{-101}
Frail	10152	1019	2.98 (2.77-3.21)	2.53×10^{-185}
Age group				
Middle age	258912	8396	1 (Ref)	

Older age	60224	2112	0.91 (0.86-0.96)	3.65×10^{-4}
Sex				
Females	164679	6368	1 (Ref)	
Males	154457	4140	0.73 (0.70-0.76)	3.64×10^{-54}
Race				
Ethnic minorities	15820	444	1 (Ref)	
White	303316	10064	1.53 (1.39-1.69)	3.54×10^{-17}
Deprivation				
Higher	97026	4086	1 (Ref)	
Middle	110046	3369	0.84 (0.80-0.88)	2.32×10^{-12}
Lower	112064	3053	0.81 (0.77-0.85)	1.67×10^{-15}
Family average Income				
Unknown	29563	964	1 (Ref)	
Low (<£51999)	210785	7909	1.15 (1.07-1.23)	6.40×10^{-5}
Middle (£52000-£100k)	61983	1406	0.92 (0.84-1.00)	0.0494
High (>£100k)	16805	229	0.59 (0.51-0.68)	1.71×10^{-12}
Education				
Less than college	211705	7794	1 (Ref)	
Above College	107431	2714	0.89 (0.85-0.93)	4.37×10^{-7}
Smoking status				
Never	176059	4955	1 (Ref)	
Ever	143077	5553	1.32 (1.27-1.37)	6.9×10^{-42}
Alcohol intake frequency				
Daily or almost daily	68444	1946	1 (Ref)	
3-4 times a week	77514	2035	0.93 (0.88-0.99)	0.0315
1-2 times a week	83261	2607	1.00 (0.95-1.07)	0.904
1-3 times a month	35130	1360	1.14 (1.06-1.22)	3.68×10^{-4}
Special occasions only	33214	1508	1.17 (1.09-1.26)	8.20×10^{-6}
Never	21573	1052	1.3 (1.2-1.4)	1.11×10^{-10}
Sedentary behavior				
0-4 hours/day	230365	6806	1 (Ref)	
>4 hours/day	88771	3702	1.07 (1.03-1.12)	1.09×10^{-3}
Metabolic syndrome				
No	229967	6711	1 (Ref)	
Yes	89169	3797	1.18 (1.13-1.23)	7.56×10^{-15}
Sleep				
7-9 h/day	237832	6961	1 (Ref)	
>9 h/day	4274	304	1.66 (1.48-1.86)	1.69×10^{-17}
<7 h/day	77030	3243	1.28 (1.23-1.33)	4.72×10^{-30}
Cancer diagnosis				
No	295142	9559	1 (Ref)	
Yes	23994	949	1.07 (1.00-1.15)	0.0361

Household size				
One person	55424	2425	1 (Ref)	
Two people	151955	4765	0.85 (0.81-0.89)	1.53×10 ⁻¹⁰
>=3 people	111757	3318	0.93 (0.88-0.99)	0.0173
Employment status				
Not employed	126349	4849	1 (Ref)	
Employed	192787	5659	0.92 (0.87-0.96)	1.43×10 ⁻⁴
Air pollution				
Low	110438	3221	1 (Ref)	
Middle	105013	3378	0.99 (0.95-1.04)	0.804
High	103685	3909	1.05 (1.00-1.11)	0.0493

3. In the results section, the authors found that "frail and pre-frail participants are more likely to smoke and spend more time in sedentary behaviors but drink alcohol less frequently than non-frail individuals." This is an intriguing result, suggesting that alcohol consumption may protect individuals from frailty. Could the authors provide further explanation and discussion on this finding?

We thank the reviewer for bringing this important issue to our attention. The counterintuitive finding that frail people drink less frequently than non-frail individuals can be explained by a reverse causation, with those at poor health are likely to reduce or quit alcohol consumption. It can also be explained by abstainer bias, where the non-drinking group often includes former drinkers who have given up alcohol because of health problems^{8,9}.

- a. To validate the above hypothesis, we included the alcohol drinker status (which categorized participants into never, previous, and current drinkers, Field ID: 20117) and the reason for former drinkers stopped drinking alcohol (Field 3859) in the current study, and found that:
 - 44.73% (10735/24000) participants who had an alcohol intake frequency of 'Never' were indeed former drinkers.
 - Among former drinkers, 47.67% (5117/10735) explicitly indicated that they stopped drinking alcohol because they have poor health or have been advised not to (Figure shown below). About 50% participants did not provide an explicit reason for stopping drinking alcohol.
- b. In examining the prospective association between frailty status and depression incidence, we used alcohol drinker status rather than alcohol intake frequency as a covariate and found nearly unchanged results. Specifically, the risk was 1.61-times (HR=1.61, 95% CI=[1.55, 1.68], 4.12×10⁻¹²²) higher for pre-frailty and

3.28-times (HR=3.28, 95% CI=[3.06, 3.51], 2.91×10^{-243}) higher for frailty, compared with non-frail individuals.

- c. Regarding the prospective association between alcohol intake status and depression, we found a 1.39-times (HR=1.39, 95% CI=[1.23, 1.57], 6.49×10^{-8}) higher risk for former drinkers compared with never drinkers; while no significant difference was observed between never drinkers and current drinkers.

These results add further evidence that former drinkers may reduce or stop drinking alcohol due to poor health. We have added the above results to a new Table S14. In the main text, we added to the discussion as follows:

Page 14: “Perhaps counterintuitive, our finding indicated that frail and pre-frail participants drank alcohol less frequently than non-frail individuals. This result can be explained by abstainer bias, where the non-drinking group often includes former drinkers who have given up alcohol because of health problems^{8, 9}. Our additional analyses suggest that former drinkers had a significantly increased risk for depression than those who never drink or current drinkers (Table S14).”

Figure in Response. Reason for former drinkers stopped drinking alcohol

Table S14. Association of alcohol status and frailty status with depression

Alcohol status	Number of subjects (percentage)			Prospective association with depression	
	Non-frail	Pre-frail	Frail	HR (95% CI)	P
Total N	202916	138093	11231		

Never	5612 (2.77%)	6575 (4.76%)	1078 (9.60%)	1 (Ref)	
Previous	4466 (2.20%)	5233 (3.79%)	1036 (9.22%)	1.39 (1.23-1.57)	6.49×10 ⁻⁸
Current	192838 (95.03%)	126285 (91.45%)	9117 (81.17%)	0.94 (0.86-1.04)	0.218

Among former drinkers, 47.67% (5117/10735) explicitly indicated that they stopped drinking alcohol because they have poor health or have been advised not to. In examining the prospective association between frailty status and depression incidence, we included alcohol drinker status rather than alcohol intake frequency and found nearly unchanged results. Specifically, the risk was 1.61-times (HR=1.61, 95% CI=[1.55, 1.68], 4.12×10^{-122}) higher for pre-frailty and 3.28-times (HR=3.28, 95% CI=[3.06, 3.51], 2.91×10^{-243}) higher for frailty, compared with non-frail individuals. Regarding the prospective association between alcohol intake status and depression, we found a 1.39-times (HR=1.39, 95% CI=[1.23, 1.57], 6.49×10^{-8}) higher risk for former drinkers compared with never drinkers, while no significant difference was observed between never drinkers and current drinkers.

4. In the results section, the authors used PHQ-9 scores to assess the association between depressive symptoms and regional GMV. Could the authors clarify why they did not divide the sample into depression cases and controls for the association analysis?

We apologize for the lack of clarity. The baseline assessment was achieved between 2006 and 2010 and followed up for about 12.25 years, while the neuroimaging data were collected after a mean of 8.87 years relative to the baseline visit (since 2014). The primary reason for using PHQ-9 scores to assess depressive symptoms in the neuroimaging analysis is that only a small number of participants (about 2.60%) were diagnosed with depression at that time. Performing analysis using considerably unmatched samples between groups (2.60% vs 97.4%) is liable to generate unreliable results. In response to the reviewer, we have clarified it in the main text as follows:

Page 15: “Symptoms of depression were measured at the imaging assessment using the 9-item version of the Patient Health Questionnaire (PHQ-9) rather than clinical depression diagnosis due to a limited number of participants diagnosed with depression when the imaging was performed (about 2.60%).”

5. Could the authors explain the rationale behind selecting the 9 inflammatory markers?

We apologize for the lack of clarity. The inclusion of these 9 inflammatory markers was based on their sensitivity to different inflammatory processes and the availability in the UK Biobank. Indeed, these 9 inflammatory markers are among the most used markers by other studies. For example, a recent study investigating the mediating effect of inflammation on the association between adiposity and dementia used the same set of inflammatory markers¹⁰. In response to the reviewer, we have added the rationale behind

selecting these inflammatory markers in the main text. We also acknowledged the test of other inflammatory markers as a potential limitation as follows:

Page 16: “Based on their sensitivity to different inflammatory processes and the availability in the UK Biobank, nine inflammatory markers were included in this study.”

Page 13: “In addition, the current study examined the mediating effect of nine inflammatory markers. Inflammatory markers reflecting more specific depression-related pathways warrant further inquiry¹¹.”

I recommend the publication of this paper once the authors address the above concerns.

We would like to thank the reviewer for the positive comments again.

References

1. Choi KW, Stein MB, Nishimi KM, *et al.* An Exposure-Wide and Mendelian Randomization Approach to Identifying Modifiable Factors for the Prevention of Depression. *The American journal of psychiatry* **177**, 944-954 (2020).
2. Yang T, Wang J, Huang J, Kelly FJ, Li G. Long-term Exposure to Multiple Ambient Air Pollutants and Association With Incident Depression and Anxiety. *JAMA psychiatry* **80**, 305-313 (2023).
3. Watson KT, Simard JF, Henderson VW, *et al.* Incident Major Depressive Disorder Predicted by Three Measures of Insulin Resistance: A Dutch Cohort Study. *The American journal of psychiatry* **178**, 914-920 (2021).
4. Sarris J, Thomson R, Hargraves F, *et al.* Multiple lifestyle factors and depressed mood: a cross-sectional and longitudinal analysis of the UK Biobank (N = 84,860). *BMC Med* **18**, 354 (2020).
5. Dregan A, Rayner L, Davis KAS, *et al.* Associations between depression, arterial stiffness, and metabolic syndrome among adults in the UK Biobank population study: a mediation analysis. *JAMA psychiatry* **77**, 598-606 (2020).
6. Zhao Y, Yang L, Sahakian BJ, *et al.* The brain structure, immunometabolic and genetic mechanisms underlying the association between lifestyle and depression. *Nature Mental Health* **1**, 736-750 (2023).
7. Gillies CL, Rowlands AV, Razieh C, *et al.* Association between household size and COVID-19: A UK Biobank observational study. *Journal of the Royal Society of Medicine* **115**, 138-144 (2022).

8. Stockwell T, Zhao J, Panwar S, *et al.* Do “moderate” drinkers have reduced mortality risk? A systematic review and meta-analysis of alcohol consumption and all-cause mortality. *Journal of studies on alcohol and drugs* **77**, 185-198 (2016).
9. Baumeister SE, Finger JD, Glaser S, *et al.* Alcohol consumption and cardiorespiratory fitness in five population-based studies. *Eur J Prev Cardiol* **25**, 164-172 (2018).
10. Deng YT, Li YZ, Huang SY, *et al.* Association of life course adiposity with risk of incident dementia: a prospective cohort study of 322,336 participants. *Molecular psychiatry* **27**, 3385-3395 (2022).
11. Ye Z, Kappelmann N, Moser S, *et al.* Role of inflammation in depression and anxiety: Tests for disorder specificity, linearity and potential causality of association in the UK Biobank. *EClinicalMedicine* **38**, 100992 (2021).

REVIEWERS' COMMENTS

Reviewer #1 (Remarks to the Author):

The authors have adequately responded to my suggestions.

Reviewer #2 (Remarks to the Author):

I appreciate the detailed and high-quality responses the author has made in a short period of time. The author has already answered all of my questions, I have no further questions.

Reviewer #2 (Remarks on code availability):

The code provide a README file with enough instructions